# Removal of Sulfonamide Resistance Genes in Fishery Reclamation Mining Subsidence Area by Zeolite

**DOI:** 10.3390/ijerph19074281

**Published:** 2022-04-02

**Authors:** Tao Yuan, Zi-Bo Lin, Sen Cheng, Rui Wang, Ping Lu

**Affiliations:** 1School of Architectural Decoration, Jiangsu Vocational Institute of Architectural Technology, Xuzhou 221000, China; yuantaocumt@126.com; 2School of Environment Science and Spatial Informatics, China University of Mining and Technology, Xuzhou 221000, China; lzb_9775@126.com (Z.-B.L.); chengsen9711@163.com (S.C.); ts21160093a31@cumt.edu.cn (R.W.)

**Keywords:** aquaculture wastewater, antibiotic resistance genes, microbial communities, water treatment

## Abstract

A majority of subsidence lakes were reclaimed as fish ponds, but the widespread use of antibiotics has caused the pollution of antibiotic resistance genes (ARGs). This paper uses zeolite as a filter material to construct a horizontal submersible wastewater treatment device and explores its effect on the removal of conventional pollutants and sulfonamide ARGs in wastewater. The results showed that the removal of total nitrogen and ammonia nitrogen by the zeolite filter media were 59.0% and 63.8%, respectively, which were higher than the removal of total phosphorus and COD. The absolute abundances of *sul1* and *sul2* in wastewater were 2.81 × 10^4^ copies·L^−1^ and 2.42 × 10^3^ copies·L^−1^. On average, 60.62% of *sul1* and 75.84% of *sul2* can be removed, and more than 90% of *sul1* and *sul2* can be removed. Experiments showed that the residence time of wastewater in the treatment device had a significant impact on removal. The microbial community structure of aquaculture wastewater was quite different before and after wastewater treatment. The abundance changes of *Saccharimonadales* and *Mycobacterium* affect the removal of sulfonamide ARGs.

## 1. Introduction

There are 98,834 abandoned mines in China, among which 45,490 are located in the cultivated area, with more than 630,000 hectares of damaged land (data as of the end of 2018). The mines in the Huanghuaihai Plain and the middle and lower reaches of the Yangtze River are located in high water table areas. Due to the shallow burial of the groundwater table, a large area of subsidence lakes is formed. With advanced land reclamation technology, the damaged land can be reused [1]. After mine reclamation, it is mainly used for agriculture, aquaculture, combined planting–fishing, and livestock and poultry breeding. However, aquaculture is the main source of antibiotics and antibiotic resistance genes (ARGs) in the environment, and agricultural activities such as organic fertilizer farming and sewage irrigation are also important sources of ARGs.

Previous research showed ARGs, including *sul1*, *sul2, bacA*, *tet36*, *tet39*, *evgS*, *Sav1866*, *macB*, etc., are abundantly enriched in the sediments and well water of coal mining subsidence in Eastern China [2]. Residual antibiotics can induce microbial resistance and aggravate the emergence of new pollutants of ARGs in the environment [3,4]. 

Currently, in the Baltic Sea area farms in Finland [5], Chilean freshwater trout farming area [6], Japanese mariculture area [7], Vietnamese freshwater farming area [8], China Hainan, Guangdong, Tianjin, Hangzhou, Yantai, Taihu Lake, and other regions have carried out research on the pollution of ARGs in aquaculture water bodies or fish bodies [9,10,11,12]. The abundance of various ARGs in aquaculture areas around the world is significantly higher than that in non-aquaculture areas, and the detection rate and relative abundance of sulfonamide ARGs are both higher [2,13]. Sulfonamide antibiotics are commonly used antibiotics in the aquaculture industry. They can treat diseases caused by microorganisms and are difficult to be completely absorbed. Most of them are excreted through feces and urine [14,15]. Therefore, sulfa antibiotics are frequently detected in various natural water bodies [6,16]. This situation exacerbates the proliferation of sulfonamide ARGs in the environment. Untreated or improperly treated aquaculture wastewater discharged into the environment will pollute surface water and groundwater and will eventually seriously endanger human health. The emergence of ARGs, a new type of pollutant, places higher demands on wastewater treatment technology.

Traditional water treatment processes cannot completely remove ARGs, and a large amount of ARGs still remain in the effluent [17,18]. Constructed wetlands have good treatment effects on a variety of polluted water bodies such as domestic sewage and agricultural wastewater [19,20,21,22]. In the existing investigations on wetlands, constructed wetlands can effectively remove ARGs [23], and the removal efficiency of ARGs in horizontal subsurface wetlands is higher than that of vertical subsurface wetlands [24,25]. ARGs are closely related to microorganisms. Previous studies have only reported the changes in 16S rRNA abundance or the microbial community structure in the wetland application matrix [26,27], but the changes in the microbial community structure before and after treatment are unclear. Therefore, in this paper, zeolite, which is often used as a wetland substrate, is used as the main filter material to construct a horizontal submersible wastewater treatment device. Combining the changes in the structure of the influent and effluent microbial communities, we explore the treatment effects and removal mechanisms of sulfonamide ARGs (*sul1* and *sul2*) in wastewater.

## 2. Materials and Methods

### 2.1. Experimental Device

A horizontal subsurface flow treatment device (length 60 cm, width 40 cm, height 40 cm) was built in a laboratory, as shown in Figure 1. The middle part of the device is the main filling area, which is 50 cm long, and the two ends (with the length of 5 cm) are, respectively, the water inlet and outlet areas. The middle of the device is filled with zeolite with a particle size of 4~8 mm (Henan Jingying Water Treatment Material Co., Ltd., Henan, China), the filling height is 30 cm, and the effective volume is about 30 L. The water inlet and outlet areas at the front and rear ends of the treatment device are filled with gravel with a particle size of 10~20 mm, and the filling height is 30 cm. There are partitions in the water inlet area, water outlet area, and the middle filling area, and a row of small holes with a diameter of 5 mm are arranged on the partition board at the upper part of the water inlet area and the bottom of the water outlet area to promote uniform water distribution. The upper part of the water outlet area is provided with a water outlet.

### 2.2. Source and Quality of Experimental Water

The raw water from the fish pond in the aquaculture area was used as experimental water. The aquaculture area is located in a coal mining subsided fish pond in northern Xuzhou, China. It began to collapse in the 1990s and adopted a fishery reclamation model. The fish ponds cover a total area of more than 1000 acres, and the area of a single fish pond is about 10 acres. Ponds are 2.5 m to 5 m deep, and are mostly 3 m deep. Samples were collected in August 2020.

### 2.3. Experiment Operation and Sample Collection

Set the hydraulic retention time to 3 days (d), the flow rate is about 10 L·d^−1^, and use a peristaltic pump to feed water. Using the continuous water inlet method, the device starts timing at the water outlet, and after 5 days of stable operation, samples are collected at the water inlet and outlet. A total of 30 days were used, and influent and effluent samples were collected at 5, 10, 20, and 30 days, respectively.

### 2.4. Methods

The pH value, the concentration of conventional pollutants (COD, total phosphorus (TP), total nitrogen (TN), and ammonia nitrogen), the absolute abundance of ARGs (*sul1*, *sul2*, and 16S rRNA) in incoming and outgoing water samples were detected, and a microbial diversity test was conducted.
Determination of conventional pollutants

The pH value of the sample was measured with a pH meter (PXSJ-216F, Shanghai Jingke Lei Magnetic Co., Ltd., Shanghai, China), and COD and ammonia nitrogen were measured with a multi-parameter water quality analyzer (5B-3B (V8), Beijing Lianhua Yongxing Technology Development Co., Ltd., Beijing, China). TN is determined by potassium persulfate oxidation UV spectrophotometry, and TP is determined by ammonium molybdate spectrophotometry.
2.Determination of gene abundance of ARGs and 16S rRNA

The genetic test water sample was suction filtered through a 0.22 μm filter membrane. After completion, the filter membrane was collected and placed in a sterile centrifuge tube and stored in ultra-low temperature at −80 °C. The filter membrane was sent to Shanghai Meiji Biotechnology Co., Ltd., Shanghai, China, for sequencing. FastDNA^®^ Spin Kit for Soil (MP Biomedicals, Santa Ana, CA, USA) was used to extract genomic DNA from samples. Electrophoresis 1% agarose gel (dyy-6c bistable electrophoresis instrument, Beijing Junyi Dongfang Electrophoresis Equipment Co., Ltd., Beijing, China) was used to detect the quality and integrity of DNA samples. The absolute abundance of sulfonamide ARGs (*sul1* and *sul2*) and 16S rRNA was determined by fluorescence quantitative PCR (ABI7300 type, Applied Biosystems, Foster City, CA, USA). The sequencing primers are shown in Table 1.
3.Determination of microbial diversity

Sequencing was performed by using the Illumina MiSep platform. Extract non-repetitive sequences, reduce the amount of redundant calculation in the middle of analysis, and remove single sequences without repetition; perform OTU clustering on non-repetitive sequences (excluding single sequences) according to 97% similarity, remove chimeras, and obtain the representative OTU sequences. Map the optimized sequence to the OTU representative sequence, select the sequence that is more than 97% similar to the OTU representative sequence, and generate the OTU table. The RDP classifier Bayesian algorithm was used to perform taxonomic analysis on a 97% similar level of OTU representative sequences, and the community species composition of each sample was counted.

## 3. Results

### 3.1. Removals of Conventional Pollutants

The pH of the original water sample was 7.89 ± 0.09, and the concentrations of COD was 29.46 ± 2.07 mg /L; TP was 1.94 ± 0.08 mg /L; TN and ammonia nitrogen were 6.54 ± 0.10 mg /L and 1.77 ± 0.07 mg /L. After treatment by the device, the pH value of aquaculture wastewater increased, and the pH value is between 8.31 and 8.46, with an average of 8.39. The removal of COD, TP, TN, and ammonia nitrogen by the treatment device is shown in Figure 2. During the operation period, the removal of conventional pollutants is relatively stable. The zeolite filler could better remove ammonia nitrogen, but did not remove COD as effectively. The average removal of COD, TP, TN, and ammonia nitrogen was 42.9%, 54.2%, 59.0% and 63.8%, respectively. The concentrations of COD, TP, TN, and ammonia nitrogen in the effluent were 16.81 ± 0.93 mg/L, 0.89 ± 0.04 mg/L, 2.68 ± 0.11 mg/L, and 0.64 ± 0.02 mg/L, respectively.

### 3.2. Removals of Sulfonamide ARGs

The absolute abundance of sulfonamide ARGs in aquaculture wastewater is relatively high. During the operation of the experiment, the average absolute abundance of ARGs is 2.81 × 10^4^ copies·L^−1^ and 2.42 × 10^3^ copies·L^−1^. The average absolute abundance of 16S rRNA is 1.71 × 10^7^ copies·L^−1^. The removal was calculated based on the absolute abundance of sulfonamide ARGs (*sul1* and *sul2*) and 16S rRNA, and the results are shown in Figure 3. With the increase in running time, the removal of *sul1* has a gradually increasing trend, while the removal of *sul2* and 16S rRNA first increases and then decreases. When running for 20 days, the removal of *sul2* and 16S rRNA reached the highest, which were 95.78% and 92.41%, respectively. When running for 30 days, the removal of *sul1* was the highest, which was 86.38%. On average, the removals of *sul1* and *sul2* by were 60.62% and 75.84%, respectively.

In order to further explore the removal of sulfonamide ARGs (*sul1* and *sul2*), the changes in their relative abundance (ratio to the absolute abundance of 16S rRNA) were analyzed, and the results are shown in Figure 4. It can be seen that the relative abundance of *sul1* has a increasing trend before and after removal, while the relative abundance of *sul2* has a decreasing trend. It can be seen that when the removal of *sul1* is lower than 16S rRNA, the relative abundance of *sul1* in the effluent is higher than that of the influent; when the removal of *sul1* is higher than 16S rRNA, the relative abundance of *sul1* in the effluent is lower than that of the influent. The change trend of *sul2* is the same. Results show that ARGs are closely related to 16S rRNA, and the change of microbial community structure affects the removal of ARGs.

### 3.3. Changes in Microbial Community Structure

The structure of the microbial community in the inlet and outlet water of the device is shown in Figure 5. At the phylum level, *Actinobacteria* is the most dominant phylum, accounting for 25.37% and 35.66% of influent and effluent, respectively. Secondly, the more dominant bacteria phyla in the influent water are *Patescibacteria*, *Proteobacteria*, *Bacteroidota*, and *Firmicutes*, etc. The effluent dominant bacteria phyla are *Actinobacteria*, *Proteobacteria*, *Bacteroidota*, *Cyanobacteria*, and *Patescibacteria*. After treatment with the device, the relative abundance of *Actinobacteria* and *Proteobacteria* increased, while the relative abundance of *Patescibacteria* and *Bacteroidota* decreased.

At the genus level, *norank_o_Saccharimonadales* is the most dominant genus of influent bacteria, accounting for 21.48%. Secondly, *norank_f_67-14, hgcI_clade*, *Mycobacterium*, and *CL500-29_marine_group* have higher relative abundances. After treatment, the relative abundance of *norank_o_Saccharimonadales* was greatly reduced, accounting for only 0.55% in the effluent, while the relative abundance of *Mycobacterium* increased, accounting for 19.04% in the effluent. The dominant bacteria in the effluent are *Mycobacterium*, *norank_f_67-14*, *norank_o_Vampirovibrionales*, *Flavobacterium*, and *CL500-29_marine_ group*. There is a big difference in the microbial community structure between the inlet water sample and the outlet water sample.

## 4. Analysis and Discussion

Previous studies have shown that 4~8 mm zeolite has a higher removal of TN than TP and COD [28], which is consistent with this study. Lu et al. [29] compared and analyzed the removal of 29 wetland fillers on ammonia nitrogen, and the results showed that zeolite has a good removal on ammonia nitrogen, which induces a higher ammonia nitrogen removal for the constructed wetland with a zeolite filler. Substrate adsorption, plant absorption, and microbial degradation are the main mechanisms for the removal of pollutants in constructed wetlands [24,30]. The research results in this paper show that only matrix adsorption has a certain removal effect on conventional pollutants, but it is lower than the removal by the constructed wetlands. Zhang et al. [31] studied the removal of sulfonamides ARGs from livestock and poultry farming wastewater by constructed wetlands. The average removals of *sul1* and *sul2* were 89% and 88%, respectively. Liu et al. [32] used constructed wetlands to remove ARGs and showed that the removal of sulfonamide ARGs reaches 85%~95%. The higher removal may be attributed to the role of plants in wetlands, which can increase the abundance of microorganisms in wetlands and improve the removal of pollutants [33]. This study can be attributed to the effect of matrix adsorption on the removal of ARGs. During the application and operation of the device, the removal of conventional pollutants is relatively stable, while the removal of ARGs fluctuates greatly, indicating that the removal mechanisms of ARGs and conventional pollutants are different. Most studies have shown that ARGs and 16S rRNA are significantly positively correlated [34,35]; thus, it is necessary to explore the removal mechanism at the microbial level.

It is more effective to explore the ARG’s removal mechanism from the microbial level, and the removal of ARGs mainly depends on the filtration effect of the matrix on the microorganisms [25,36,37]. In the early stage of operation, the removal of *sul1*, *sul2*, and 16S rRNA of the device was low or because the filtering effect on microorganisms was low in the early stages of operation. As the running time increases, the suspended solids and microorganisms in the water body narrow the gaps of the zeolite, thereby improving the removal of pollutants. Differences in the removal of the two sulfonamide ARGs by the treatment device may be due to differences in the properties of the genes themselves [38]. *Sul1* is usually associated with other ARGs in class 1 integrons, while *sul2* is usually located on a small non-binding plasmid or a large transmissible multi-resistance plasmid [39,40]. After aquaculture wastewater is treated by the device, the structure of the microbial community changes, which may explain the removal of sulfonamide ARGs. The relative abundance of *Patescibacteria* decreased after wastewater treatment. *Saccharimonadales* belongs to *Patescibacteria*, and its relative abundance is also greatly reduced. Previous studies have shown that *Patescibacteria* are significantly positively correlated with ARGs [41]. *Saccharimonadales* are resistant bacteria to sulfonamide antibiotics [42]. The decrease in the abundance of such bacteria is the main reason for the decrease in the abundance of ARGs.

After treatment by the device, the relative abundance of *Mycobacterium* in the aquaculture wastewater is relatively high. Previous studies have shown that *Mycobacterium* may be a potential host for ARGs [43]. This shows that the device has a low removal of some resistant bacteria. Therefore, the treatment device can reduce the absolute abundance of sulfonamide ARGs in aquaculture wastewater, but there is still a risk of increasing their relative abundance and increasing the drug’s resistance relative to the microbial community. ARGs can be divided into intracellular ARGs and extracellular ARGs [44]. Intracellular ARGs are removed with bacteria, and the removal of total ARGs is lower than that of bacteria, indicating that the device has a lower removal of extracellular ARGs. The difference between the removal and removal mechanism of intracellular and extracellular ARGs in aquaculture wastewater needs to be explored.

## 5. Conclusions

This preliminary research discusses the effect of zeolite on the removal of ARGs in fish pond wastewater and analyzes the mechanism of the removal of ARGs from the microbial level. The main conclusions are as follows:(1)Zeolite can reduce the absolute abundance of sulfonamides ARGs and as high as 90% can be removed; the average removal of *sul1* is lower than that of *sul2*.(2)There is a big difference in the microbial community structure of the influent and effluent. The zeolite treatment affects the microbial community structure of the effluent, and the change of the microbial structure indirectly affects the removal of ARGs.(3)The experimental results found that the device can only remove part of the drug-resistant bacteria, so there is still a risk of aggravating the drug resistance of the bacterial community.

The removal of ARGs in fish ponds should be further strengthened, and field tests and demonstration engineering studies should be conducted to provide technical guidance for the long-term control of the diffusion and migration of ARGs.

## Figures and Tables

**Figure 1 ijerph-19-04281-f001:**
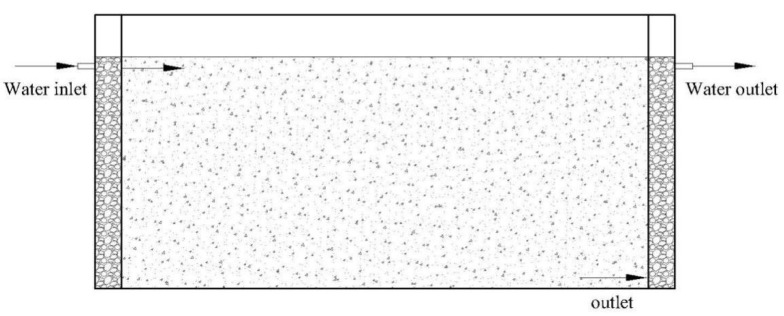
Schematic diagram of wastewater treatment device.

**Figure 2 ijerph-19-04281-f002:**
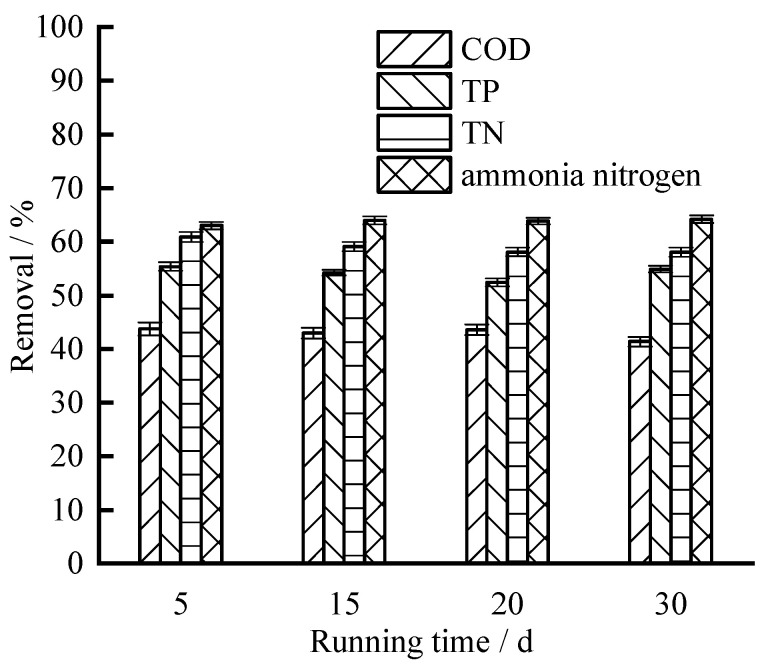
Removals of Conventional pollutants.

**Figure 3 ijerph-19-04281-f003:**
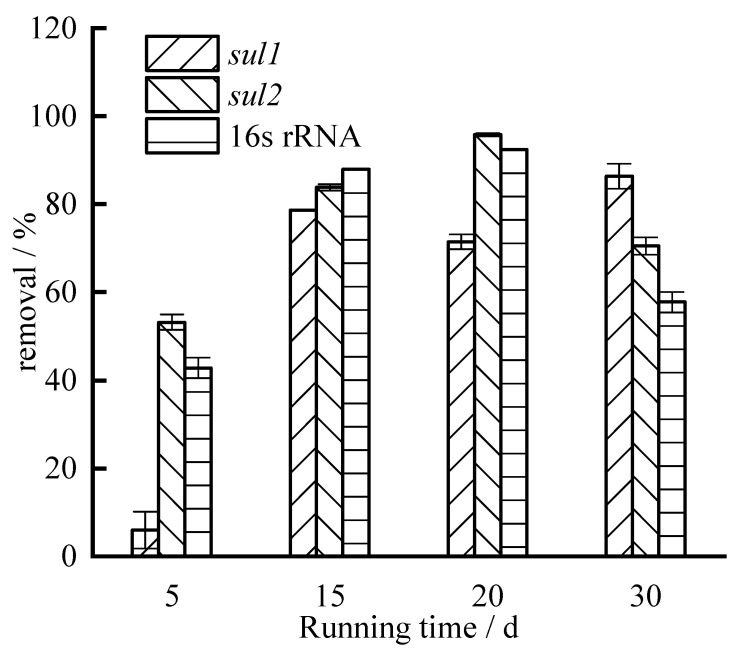
Removal of ARGs.

**Figure 4 ijerph-19-04281-f004:**
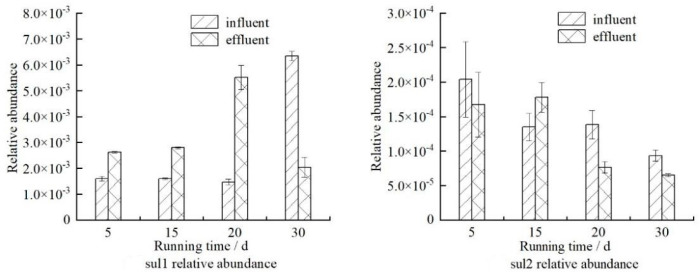
Relative abundance of sulfonamide ARGs.

**Figure 5 ijerph-19-04281-f005:**
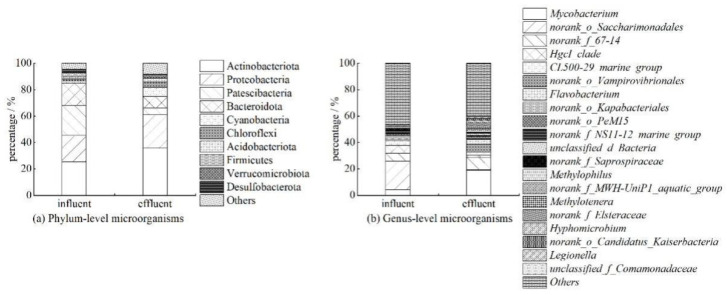
Microbial community structure in the influent and effluent of the device.

**Table 1 ijerph-19-04281-t001:** Gene sequencing primers.

Gene Name	Primer Name	Primer Sequence (5′ → 3′)	Product Size/bp	Annealing Temperature/°C
*sul1*	*sul1*-F	CACCGGAAACATCGCTGCA	159	58
	*sul1*-R	AAGTTCCGCCGCAAGGCT		
*sul2*	*sul2*-F	CTCCGATGGAGGCCGGTAT	191	58
	*sul2*-R	GGGAATGCCATCTGCCTTGA		
16S rRNA	16S-F	TGTGTAGCGGTGAAATGCG	140	62
	16S-R	CATCGTTTACGGCGTGGAC		

## Data Availability

Not applicable.

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
