# Peer review of "Removal of Sulfonamide Resistance Genes in Fishery Reclamation Mining Subsidence Area by Zeolite"

_ijerph, 2022, doi:10.3390/ijerph19074281_

Round 1

Reviewer 1 Report

This study attempted to remove the sulfonamide genes in fishery reclamation mining subsidence area, the attempted methods are fine and the results claims that the zeolite can reduce sulfonamides. However, the strong evidence of showing that the abundance changes of Saccharimonadales and Mycobacterium affect the removal effect of sulfonamide ARGs are missing. 

Few points:

In the abstract, line 19 -20  2.81×104 copies·L-1 19 and 2.42×103 - superscripts should be properly given.

In many places the bacterial names are not italizised and Language editing must be improved.

Overall,the paper can be improved for language editing and can be published

Author Response

Dear Editor and Reviewers,

Really appreciate your valuable comments and time on this manuscript. We have made revisions using the “Track Changes” function according to the referees’ comments and upload the revised file to the system. 

Please see the attachment and find the responses to comments .

Thank you again for your suggestions.

All the bests,

Authors

Reviewer 2 Report

The study entitled „Removal of Sulfonamide Resistance Genes in Fishery Reclamation Mining Subsidence Area by Zeolite” presents the recent findings on the antibiotics resistance genes distribution. In my opinion, the work is innovative and the results can be important from the practical viewpoint. However, this is a quite new topic which deserves further studies.

In my opinion, the authors used a reliable methodology and I did not find any shortcomings. The research results are presented clearly. The discussion of the results is logical and I fully agree with the conclusions made by the authors. Therefore, I believe that the manuscript does not require any improvements and can be accepted in the present form.

Author Response

Dear Editor and Reviewers,

Really appreciate your valuable comments and time on this manuscript. We have made revisions using the “Track Changes” function according to the referees’ comments and upload the revised file to the system. 

Thank you again for your suggestions.

All the bests,

Authors

Reviewer 3 Report

The paper is interesting, however it seems to be a little bit chaotic. The authors declare that it concerns the removal of sulfonamide resistance genes, but they also heve included other parameters. That is important to support the results concerning gene removal, but the text of the paper is structuralised in the way, which not present the genes removal above all. The authors are requested for including the methodology of COD, TP, TN and ammonia nitrogen analysis, because they present the results, but no details on analytical methods are given. In the part concerning the description of result I suggest to start with removal effects of genes, and the data concerning removal of conventional pollutants should be presented at the end of the paper as supporting material only. The experimental methods were choosen properly. The discussion of the results is rather short, but informative. I suggest to emhasise that the results seem to be rather preliminary, because the scope of research work was not designed as complex research of the phenomenon.

Author Response

(The authors gave the same response as above.)
